# Associations of Body Mass Index and Waist Circumference with the Risk of Head and Neck Cancer: A National Population-Based Study

**DOI:** 10.3390/cancers14163880

**Published:** 2022-08-11

**Authors:** Choung-Soo Kim, Jun-Ook Park, Inn-Chul Nam, Sung Joon Park, Dong-Hyun Lee, Hyun-Bum Kim, Kyung-Do Han, Young-Hoon Joo

**Affiliations:** 1Department of Otolaryngology-Head and Neck Surgery, College of Medicine, The Catholic University of Korea, Seoul 07345, Korea; 2Department of Otorhinolaryngology-Head and Neck Surgery, Chung-Ang University College of Medicine, Chung-Ang University Gwangmyeong Hospital, Gwangmyeon-si 14353, Korea; 3Department of Statistics and Actuarial Science, Soongsil University, Seoul 06978, Korea

**Keywords:** head and neck neoplasms, body mass index, waist circumference, epidemiology, Korea

## Abstract

**Simple Summary:**

A high body mass index is positively associated with multiple cancer types. However, there are controversies regarding such an association with head and neck cancer. This population-based study is the first report of association of body mass index and waist circumference with the risk of head and neck cancer. Low body mass index and low waist circumference were related to a risk of head and neck cancer. These findings highlight the importance of preventing and reversing body mass index to reduce head and neck cancer incidence.

**Abstract:**

**Background:** We investigated the association between BMI and HNC subtype incidence in a cohort study of ten million people, adjusting for the effect of smoking and drinking. We also investigated the relationship between waist circumference (WC) and HNC subtype. **Methods:** All data used in this study originated from the Korean National Health Insurance Service database. We analysed subjects who had undergone health check-ups in 2009 and monitored subjects until 2018 (n = 10,585,852). Finally, 9,598,085 subjects were included after exclusions. We collected variables that could affect the risk of HNC. Cox proportional hazards regression analysis was used to estimate hazard ratios (HRs) and 95% confidence intervals (CIs). **Results:** The overall incidence of HNC was higher in the low BMI category (BMI < 18.5 according to WHO recommendations for Asian people) (HR: 1.322; 95% CI: 1.195–1.464) compared with the normal BMI category. Among the HNC cases, the incidence rates of laryngeal (HR: 1.3; 95% CI: 1.085–1.558), oral cavity (HR: 1.277; 95% CI: 1.011–1.611), and oropharyngeal (HR: 1.584; 95% CI: 1.25–2.008) cancers were higher in the low BMI category compared with the normal BMI category. No significant association was detected between low BMI and sinus cancer, salivary gland cancer, or nasopharyngeal cancer. The low WC category (<80 cm in men and <75 cm in women) was related to a risk of hypopharyngeal (HR: 1.268; 95% CI: 1.061–1.514) and laryngeal (HR: 1.118; 95% CI: 1.007–1.241) cancers. The HR for occurrence of HNC was high in underweight participants according to smoking status (1.219 for never smoker vs. 1.448 for ever smoker, *p* for interaction = 0.0015) and drinking status (1.193 for never drinker vs. 1.448 for ever drinker, *p* for interaction = 0.0044). **Conclusions:** Low BMI was associated with the risk of some types of HNC. The results of this study could assist etiological investigations and prevention strategies.

## 1. Introduction

Body mass index (BMI) is the most common indicator of general obesity, and waist circumference (WC) is an indicator of abdominal obesity. Previous studies have reported that an increase in BMI is positively associated with multiple cancer types, such as colon and breast cancers [1,2,3]. However, the relationship between BMI and the incidence of head and neck cancer (HNC) remains controversial [4,5,6,7].

According to global cancer statistics, in 2020 HNC was the seventh most common type of cancer, and 931,931 new cases of HNC were diagnosed. This includes oral cavity, nasopharyngeal, oropharyngeal, hypopharyngeal, laryngeal, and salivary gland cancers [8]. Smoking and alcohol are major risk factors for HNC [9]. Some subtypes of HNC caused by smoking and alcohol can affect BMI by causing dysphagia [5]. In most studies, HNC is associated with a lower BMI compared with a normal and higher BMI [10]. After analyzing the data of 17 case–control studies that included 12,716 cases and 17,439 controls, Gaudet et al. suggested that a low BMI was associated with a higher risk of HNC, regardless of smoking and drinking [5]. However, the same author reported no association between BMI and the incidence of HNC in a prospective cohort study, except in smokers. BMI is inversely associated with HNC mortality [7]. These conflicting results originated from smaller studies with relatively short follow-up times. Furthermore, the lack of statistical power limited a stratified analysis that included smoking and drinking [7].

Due to these limitations, few studies have evaluated the association between HNC and BMI and the incidence of HNC such as laryngeal, pharyngeal, oral cavity, or salivary gland cancers. A recent study reported that central obesity, expressed as WC, is more predictive than BMI in identify the risk for cancer [11]. In this study, we investigated the association between BMI and the incidence of HNC subtypes and also the relationship between WC and the HNC subtype in a cohort study of ten million people. Further, we performed stratified analysis about smoking and drinking in the occurrence of HNC according to BMI and WC category in a cohort study of ten million people.

## 2. Materials and Methods

### 2.1. Study Population

All data used in this study came from the Korean National Health Insurance Service (NHIS) database. The NHIS is the obligatory health insurance for about 97% of the Korean population except for Medicaid beneficiaries. The NHIS database includes patient demographics and the records of diagnosis, interventions, and prescriptions. The NHIS health check-up programs include anthropometric measurements, hearing and visual acuity checks, laboratory tests, past family, medical, and surgical history, and social history. Hospitals perform the health check-ups after being certified by the NHIS, which also regularly qualifies trained examiners. Past medical history, and health-related behaviours such as smoking, alcohol intake, and physical activity were collected using standardized self-reporting questionnaires. Therefore, the Korean NIHS data represent the entire Korean population without selection bias and have been used in many epidemiological studies.

### 2.2. Patient Selection and Cancer Ascertainment

We selected subjects who were >20 years and who had undergone a health check-up in 2009 (n = 10,585,852). We monitored subjects until 31 December 2018. Participants were censored if they (1) died, (2) immigrated and were no longer followed up by the NHIS, or (3) had new-onset HNC (primary outcome of the study). We excluded individuals with missing data (n = 746,403) and those with a history of another cancer before the health check-up (n = 153,456). We also applied a 1-year lag period to minimize detection bias. Subjects who diagnosed with cancer within one year of the health check-up were excluded to minimize the effect of cancer on BMI or WC, and subjects who had less than a year of follow-up data were also excluded (n = 87,908). Finally, 9,598,085 subjects were included in this study from baseline to the date of HNC diagnosis. Participants were diagnosed with HNC if they had admission records for HNC in their NHIS data from 2010 to 2018. Diagnoses were confirmed by the International Classification of Disease, Tenth Revision, Clinical Modification codes: C02, C03, C04, C05, and C06 for oral cavity cancer; C07 and C08 for salivary gland cancer; C11 for nasopharyngeal cancer; C01, C051, C099, and C103 for oropharyngeal cancer; C12 and C13 for hypopharyngeal cancer; C10 for sinus cancer; and C32 for laryngeal cancer.

### 2.3. Definitions of BMI and WC

BMI was calculated as the weight divided by the square of the height (kg/m^2^) and categorized using the WHO recommendation for Asian populations, (underweight ≤ 18.5, normal weight = 18.5–23, overweight = 23–25 kg/m^2^, preobesity = 25–30 kg/m^2^, obesity ≥ 30 kg/m^2^) [12]. Abdominal obesity was defined as WC ≥ 90 cm in men and ≥85 cm in women, according to the definition of the Korean Society for the Study of Obesity. For analysis, WC was separated into levels of 5 cm increments: level 1, <80 cm in men and <75 cm in women; level 2, 80–85 cm in men and 75–80 cm in women; level 3, 85–90 cm in men and 80–85 cm in women; level 4, 90–95 cm in men and 85–90 cm in women; level 5, 95–100 cm in men and 90–95 cm in women; and level 6, ≥100 cm in men and ≥95 cm in women [13].

We categorized smoking status as non-smokers, ex-smokers, or current smokers. Alcohol drinking was categorized as non-drinking, mild (<30 g/day), and heavy (≥30 g/day), and regular exercise was defined as vigorous physical activity for at least 20 min/day. Income levels were divided by quartiles: Q1 (the lowest), Q2, Q3, and Q4 (the highest). Subjects with hypertension were defined as having at least one claim per year for the prescription of anti-hypertension medication under ICD-10 codes I10–13 and I15, or having a systolic blood pressure ≥ 140 mmHg or diastolic blood pressure ≥90 mmHg without a claim for anti-hypertension medication under ICD-10 codes I10–13 and I15. Diabetes was defined as a fasting blood glucose level ≥ 126 mg/dL (≥7 mmol/L) or the presence of one or more claims per year for antihyperglycemic medications with ICD-10-CM code E10-14. Patients with a glomerular filtration rate < 60 mL/min/1.73 m^2^ at the time of baseline evaluation were classified as the chronic kidney disease group.

### 2.4. Statistical Analysis

When death or cancer occurred within one year after the medical examination, subjects were excluded from the analysis to reduce bias. Cox proportional hazards regression analysis (follow-up duration is the primary time variable) was used to estimate hazard ratios (HRs) and 95% confidence intervals (CIs) for the association between BMI and the risk of HNC. Model 1 was not adjusted; Model 2 was adjusted for age, sex, income, smoking, alcohol consumption, exercise, and chronic diseases that could affect BMI or WC, such as hypertension and diabetes. In Model 2, variables that could affect the incidence of HNC or influence of BMI and WC were adjusted for. Tests for multiplicative interactions were performed for BMI category and smoking and drinking. Statistical analyses were performed using SAS ver. 9.4 (SAS Institute, Cary, NC, USA). A *p*-value < 0.05 was considered significant.

## 3. Results

### 3.1. Basic Characteristics

The general characteristics of the participants according to the BMI WHO recommendations for Asian populations are presented in Table 1. In total, 10,585,852 participants were recruited, and 987,767 participants were excluded. Finally, 9,598,085 participants were eligible for this study from 2009 to 2019. The median follow-up duration was 8.31 (8.12–8.57) years. Among them, 10,732 participants were newly diagnosed with HNC: 2972 with laryngeal, 2225 with oral cavity, 1814 with oropharyngeal, 929 with hypopharyngeal, 1101 with nasopharyngeal, 539 with sinus, and 1273 with salivary gland. The mean BMI was 17.57 ± 0.78 kg/m^2^ in the underweight group, 21.13 ± 1.21 kg/m^2^ in the normal weight group, 23.93 ± 0.57 kg/m^2^ in the overweight group, 26.73 ± 1.29 kg/m^2^ in the preobesity group, and 32 ± 6.89 kg/m^2^ in the obesity group. Participants were more likely to be non-smokers than smokers or ex-smokers. However, the ratio of current smoker was highest in the obesity group compared with other BMI categories. Drinking was also distributed in the order of non-alcohol, mild, and heavy in all BMI categories, similar to smoking. The ratio of heavy drinkers was highest in obesity group compared with other BMI categories. WC and weight were higher with higher BMI categories. The correlation of BMI and WC was 0.705 (0.700 for male and 0.730 for female).

### 3.2. Relationship between BMI and HNC

The incidence of HNC according to the BMI category is shown in Table 2. The overall incidence of HNC was higher in the low BMI category (HR: 1.322; 95% CI: 1.195–1.464) compared with the normal BMI category. Among the HNCs, the incidence rates of laryngeal (HR: 1.3; 95% CI: 1.085–1.558), oral cavity (HR: 1.277; 95% CI: 1.011–1.611), and oropharyngeal (HR: 1.584; 95% CI: 1.25–2.008) cancers were higher in the low BMI category compared with the normal BMI category. However, differences were not observed between cancer incidence and a low BMI in sinus (HR: 1.249; 95% CI: 0.769–2.026), salivary (HR: 1.26; 95% CI: 0.918–1731), hypopharyngeal (HR: 1.292; 95% CI: 0.966–1.729) and nasopharyngeal cancers (HR: 0.917; 95% CI: 0.616–1.364). Participants with a high BMI showed strong association with a lower risk of HNC. The HR of HNC incidence was 0.89 (95% CI: 0.848–0.935) in the overweight group, 0.836 (95% CI: 0.797–0.876) in the preobesity group, and 0.868 (95% CI: 0.772–0.976) in the obesity group. A higher BMI was significantly associated with a low risk of hypopharyngeal cancer (HR: 0.68; 95% CI: 0.58–0.797 in the overweight group, HR: 0.445; 95% CI: 0.373–0.532 in the preobesity group, and HR: 0.247; 95% CI: 0.122–0.498 in the obesity group) and laryngeal cancer (HR: 0.784; 95% CI: 0.715–0.86 in the preobesity group). A difference in HR among BMI categories according to HNC subsite was evaluated in laryngeal (*p* < 0.0001), sinus (*p* = 0.7536), hypopharyngeal (*p* < 0.0001), oropharyngeal (*p* = 0.0002), oral cavity (*p* < 0.0637), and nasopharyngeal (*p* = 0.8224) cancers. The HRs of the HNC subtypes according to the five BMI categories compared with normal BMI are shown in Figure 1. In all subtypes of HNC, except nasopharyngeal cancer, a low BMI had a higher HR compared with a normal BMI, regardless of significance.

### 3.3. Relationship between WC and HNC

The incidence rates of the HNC subtypes according to the WC categories are shown in Table 3. The overall incidence of HNC was higher for level I WC (HR: 1.104; 95% CI: 1.045–1.165) compared with the normal (level III) WC category. Among the HNCs, only the incidence rates of hypopharyngeal cancer (HR: 1.268; 95% CI: 1.061–1.514) and laryngeal cancer (HR: 1.118; 95% CI: 1.007–1.241) were higher in the level I WC category compared with the normal WC category.

### 3.4. Interactions of BMI with Smoking and Drinking on Occurrence of HNC

We examined the effect of smoking or drinking status on the risk of HNC according to BMI category. Table 4 provides HRs and 95% CIs among participants with ever smoker as compared with never smoker stratified by BMI. There was a positive interaction between BMI and smoking on occurrence of HNC (HR: 1.129; 95% CI: 1.024–1.451 for never smoker and underweight, HR: 1.335; 95% CI: 1.178–1.512 for ever smoker and underweight, *p* for interaction = 0.0015).

The HR for occurrence of HNC was high in underweight participants according to never or ever drinker (1.193 vs. 1.448, *p* for interaction = 0.0044) (Table 5).

## 4. Discussion

We observed an association between BMI and the incidence of HNC in this large cohort study. However, this association differed by cancer subtype. The incidence of laryngeal, oral cavity, and oropharyngeal cancer was higher in the low BMI category compared with the normal BMI category. Furthermore, the incidence of hypopharyngeal cancer was negatively correlated with higher BMI. These results were adjusted for all of the covariates listed in Table 1.

The results of our study are consistent with previous studies, as low BMI was associated with a higher risk of HNC. However, there were some differences in the HNC subtype risk analysis. The authors of the prospective NIH AARP cohort study showed an inverse association between HNC and BMI that was almost exclusively among current smokers (HR: 0.76) [14]. In a Dutch cohort study, oral cavity and oropharyngeal cancers had the strongest inverse association with BMI [6]. Our results confirm previous findings that oral, oropharyngeal, and laryngeal cancers are associated with a low BMI; however, the inverse association between BMI and oral cavity cancer failed to reach statistical significance (*p* = 0.0637). Our results suggest that the relationship between the risk of sinus, salivary gland, and nasopharyngeal cancers differed from that of other HNCs.

Drinking and smoking are known risk factors for HNC; however, the effects of drinking and smoking on BMI or weight differ among studies. In their prospective study, Gaudet et al. reported that moderate to heavy drinkers tend to have a lower BMI and a higher risk for HNC [7]. However, Traversy et al. suggested that heavy drinking is more consistently related to weight gain [15]. In this study heavy drinking was positively related to a high BMI (Table 1). A previous study proposed that smoking affects low BMI. Piirtola et al. suggested that current smoking is associated with a lower BMI and that stopping smoking is associated with a higher BMI in their monozygotic twin study [16]. In the present study, the relationship between current smoking and BMI was inconsistent, as in previous studies, as the ratio of current smokers was higher with higher BMI. Thus, smoking status and alcohol consumption should be adjusted to analyse the relationship between the incidence of HNC and BMI. However, there was limited power in most previous studies to stratify smoking status or alcohol consumption, or both, due to the numbers of participants and the study durations.

There has been no exact explanation about the relationship between BMI and HNC risk [10]. Most studies have explained that smoking and drinking can be associated with being underweight or dysphagia, or that odynophagia could influence being underweight [5]. Some recent studies suggested that neuropeptides such as leptin were influenced by smoking and drinking and affect body weight by regulating satiety and food intake [17,18,19,20,21]. Gallina reported that leptin was associated with recurrence of malignancy in laryngeal cancer [21]. Here, we adjusted the results for the effects of smoking and drinking. Participants diagnosed within one year after determining their BMI were excluded to rule out the effect of cancer on BMI. Furthermore, most laryngeal cancer cases did not develop dysphagia. A possible explanation is that patients with a low BMI have low levels of vitamins or micronutrients such as vitamins A, C, and D, or folate [22,23,24,25,26]. These vitamins and micronutrients could act as antioxidants, which are important in the prevention of further injury or carcinogenesis during eating or phonation in the upper aerodigestive tract [27,28,29,30,31,32,33,34]. Nasopharyngeal cancer might have a different pattern from the other pharyngeal cancers in this study because it rarely occurs as a result of exposure to foreign substances [35].

Some studies have associated WC with a higher risk of HNC, but we observed a relative inverse but non-significant relationship between HNC and WC. Abdominal obesity, rather than general obesity, is considered to be one of the key features of metabolic syndrome [36]. Intra-abdominal fat has been hypothesised to be biologically different from fat in other areas with regard to tumour angiogenesis and cell proliferation [37]. These results were supported by a prospective cohort study of 3.5 million adults in Spain. In that study, the HR of laryngeal cancer differed according to BMI and WC, unlike other cancer types [38]. This difference may be due to the inconsistency between BMI and WC. In the present study, most of level 3 WC was comprised of preobesity or those that were overweight according to BMI category, and a normal BMI occupied 72.77% of level 1 WC. Therefore, WC was not a good reflection of a low BMI.

The strengths of our study are that the results were analysed in cohort data including more than 9.5 million participants over 10 years. Therefore, we investigated the association between BMI and HNC subtype with sufficient power after adjusting for smoking and drinking to minimize confounding bias. The NIHS database provided the health information for 97% of the Korean population. Therefore, although data represented only Koreans, selection bias was limited. Furthermore, the hospitals that performed the health check-ups were certified by the NHIS, which also qualifies trained examiners. These factors minimized the information bias. Another strength was excluding participants who were diagnosed with HNC within 1 year of the study to rule out the effects on diet or BMI of the patient. Finally, this study compared the effect of BMI and WC on the risk of the HNC subtype.

This study also had limitations. First, most of the participants were Korean, so it was difficult to assess the representativeness of races or ethnicity. Second, the effects of HPV on oropharyngeal cancer were not included. The causes and clinical manifestations of HPV-related oropharyngeal cancer are different from those of HPV-negative oropharyngeal cancer [39,40,41,42]. However, because the diagnosis in the Korea NHIS is based on the International Classification of Diseases, we could not distinguish between oropharynx cancer related to HPV. Third, the environmental and occupational exposure of the individuals could not be considered in this study. Although there are some case–control studies about the association between nutrients and HNC, this study did not contain detailed individual nutritional status or diet. Nutritional intake is one of the leading contributing factors to obesity. Future studies should include a more accurate and comprehensive dietary assessment to determine the role of diet and body composition in HNC development [25,43,44]. Finally, this study was designed as a 10 year retrospective cohort study. The lifestyle (including smoking or drinking status) of participants could be changed during the observation period; however, it is difficult to reflect these changes. Also, the causal relationship between changes in BMI, WC, and HNC risk is beyond the scope of this cohort study. Further, well-designed case–control studies are needed.

## 5. Conclusions

In this large cohort study, low BMI was associated with a higher risk of some types of HNC, confirming the results of previous studies. This study provides the novel finding that salivary gland, sinus, and nasopharyngeal cancers have different patterns from those of other HNCs in their relationships with BMI. Unlike BMI, WC was only related to the incidence of hypopharyngeal and laryngeal cancers.

## Figures and Tables

**Figure 1 cancers-14-03880-f001:**
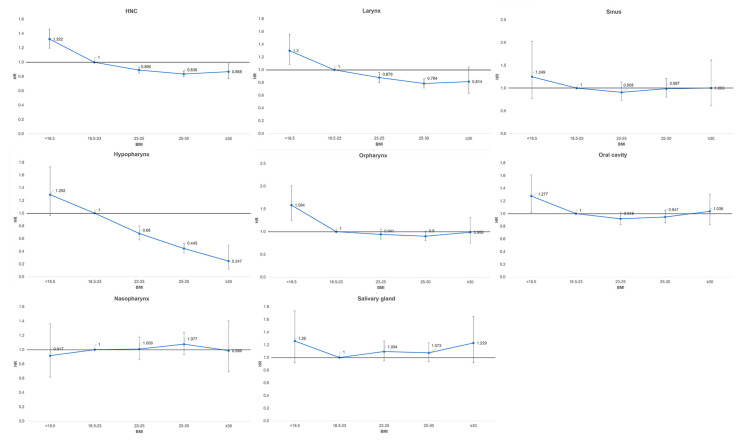
Hazard ratios (HR) of head and neck cancer (HNC) and its subtypes according to body mass index (BMI). All HRs were adjusted for age, sex, income, smoking, alcohol consumption, regular exercise, diabetes, and hypertension.

**Table 1 cancers-14-03880-t001:** General characteristics of the participants according to WHO recommendation of body mass index for the Asian population (underweight ≤ 18.5 kg/m^2^, normal weight = 18.5–23 kg/m^2^, overweight = 23–25 kg/m^2^, preobesity = 25–30 kg/m^2^, obesity ≥ 30 kg/m^2^).

	Body Mass Index Category				
	Underweight	Normal Weight	Overweight	Preobesity	Obesity
Parameter	(N = 357,470)	(N = 3,750,066)	(N = 2,358,962)	(N = 2,789,174)	(N = 342,413)
Age (years)					
<40	210,623 (58.92%)	1,380,462 (36.81%)	607,317 (25.75%)	708,390 (25.4%)	123,406 (36.04%)
40–64	102,359 (28.63%)	1,944,746 (51.86%)	1,432,649 (60.73%)	1,685,288 (60.42%)	178,026 (51.99%)
≥65	44,488 (12.45%)	424,858 (11.33%)	318,996 (13.52%)	395,496 (14.18%)	40,981 (11.97%)
Sex					
Male	116,318 (32.54%)	1,754,833 (46.79%)	1,405,704 (59.59%)	1,752,540 (62.83%)	191,406 (55.9%)
Female	241,152 (67.46%)	1,995,233 (53.21%)	953,258 (40.41%)	1,036,634 (37.17%)	151,007 (44.1%)
Smoking					
Non	254,354 (71.15%)	2,427,070 (64.72%)	1,359,999 (57.65%)	1,533,879 (54.99%)	197,060 (57.55%)
Ex	22,405 (6.27%)	397,607 (10.6%)	377,572 (16.01%)	483,365 (17.33%)	44,503 (13%)
Current	80,711 (22.58%)	925,389 (24.68%)	621,391 (26.34%)	771,930 (27.68%)	100,850 (29.45%)
Drinker					
Non	202,657 (56.69%)	2,001,560 (53.37%)	1,186,970 (50.32%)	1,370,271 (49.13%)	177,853 (51.94%)
Mild	138,731 (38.81%)	1,511,588 (40.31%)	975,536 (41.35%)	1,138,473 (40.82%)	128,245 (37.45%)
Heavy	16,082 (4.5%)	236,918 (6.32%)	196,456 (8.33%)	280,430 (10.05%)	36,315 (10.61%)
Regular Exercise	33,367 (9.33%)	605,013 (16.13%)	465,686 (19.74%)	545,871 (19.57%)	58,475 (17.08%)
Income (Q1)	80,596 (22.55%)	783,837 (20.9%)	440,059 (18.65%)	505,171 (18.11%)	68,253 (19.93%)
Diabetes	11,717 (3.28%)	205,662 (5.48%)	210,701 (8.93%)	343,044 (12.3%)	59,515 (17.38%)
Hypertension	33,310 (9.32%)	606,832 (16.18%)	630,040 (26.71%)	1,025,939 (36.78%)	170,723 (49.86%)
Age (years)	40.18 ± 16.52	45.13 ± 14.31	48.66 ± 13.31	49.1 ± 13.21	46.19 ± 13.83
Height (cm)	162.8 ± 8.15	163.33 ± 8.82	164.12 ± 9.33	164.39 ± 9.64	164.09 ± 10.6
Weight (kg)	46.69 ± 5.18	56.55 ± 7.02	64.67 ± 7.5	72.49 ± 9.15	86.42 ± 12.31
Body mass index (kg/m^2^)	17.57 ± 0.78	21.13 ± 1.21	23.93 ± 0.57	26.73 ± 1.29	32 ± 6.89
Waist circumference (cm)	66.14 ± 6.03	74.23 ± 6.8	81.2 ± 6.28	87.16 ± 6.67	96.8 ± 8.39

Note: Drinking: non-drinking, mild (<30 g/day), and heavy (≥30 g/day). Regular exercise: activity for at least 20 min/day. Income: Q1 (the lowest), Q2, Q3, and Q4 (the highest). N: number of participants.

**Table 2 cancers-14-03880-t002:** Hazard ratios of head and neck cancer and its subtypes according to body mass index level.

				Per 1000 People-Year				
Body Mass Index	N	Event	Duration	Incidence Rate	Model 1	*p*-Value	Model 2	*p*-Value
**Head and neck cancer**								
18.5	357,470	414	2,887,271	0.143	1.096 (0.990, 1.212)	<0.0001	1.322 (1.195, 1.464)	<0.0001
18.5–23	3,750,066	4040	30,818,312	0.131	1 (Ref.)		1 (Ref.)	
23–25	2,358,962	2793	19,444,233	0.143	1.095 (1.043, 1.149)		0.890 (0.848, 0.935)	
25–30	2,789,174	3174	22,993,741	0.138	1.052 (1.004, 1.102)		0.836 (0.797, 0.876)	
≥30	342,413	311	2,819,334	0.110	0.841 (0.75, 0.944)		0.868 (0.772, 0.976)	
per 1 kg/m^2^					0.998 (0.992, 1.004)	0.5708	0.969 (0.963, 0.975)	<0.0001
**Larynx cancer**								
<18.5	357,470	132	2,887,888	0.045	1.196 (0.999,1.432)	<0.0001	1.300 (1.085,1.558)	<0.0001
18.5–23	3,750,066	1179	30,826,388	0.038	1 (Ref.)		1 (Ref.)	
23–25	2,358,962	783	19,450,261	0.040	1.052 (0.961, 1.151)		0.879 (0.802, 0.963)	
25–30	2,789,174	811	23,000,953	0.035	0.921 (0.842, 1.007)		0.784 (0.715, 0.86)	
≥30	342,413	67	2,820,051	0.023	0.621 (0.485, 0.794)		0.814 (0.634, 1.044)	
per 1 kg/m^2^					0.975 (0.964, 0.986)	<0.0001	0.959 (0.947, 0.971)	<0.0001
**Sinus cancer**								
<18.5	357,470	18	2,888,192	0.006	1.013 (0.625, 1.643)	0.2755	1.249 (0.769, 2.026)	0.7536
18.5–23	3,750,066	190	30,829,643	0.006	1 (Ref.)		1 (Ref.)	
23–25	2,358,962	133	19,452,634	0.006	1.108 (0.888, 1.383)		0.908 (0.727, 1.135)	
25–30	2,789,174	179	23,003,264	0.007	1.261 (1.028, 1.546)		0.987 (0.801, 1.215)	
≥30	342,413	19	2,820,274	0.006	1.091 (0.681, 1.749)		1.003 (0.622, 1.617)	
per 1 kg/m^2^					1.002 (0.999, 1.005)	0.1836	0.993 (0.965, 1.021)	0.6149
**Hypopharynx cancer**								
<18.5	357,470	51	2,888,147	0.017	1.195 (0.894,1.596)	<0.0001	1.292 (0.966, 1.729)	<0.0001
18.5–23	3,750,066	457	30,829,108	0.014	1 (Ref.)		1(Ref.)	
23–25	2,358,962	234	19,452,430	0.012	0.810 (0.692, 0.948)		0.680 (0.580, 0.797)	
25–30	2,789,174	179	23,003,378	0.007	0.524 (0.441, 0.623)		0.445 (0.373, 0.532)	
≥30	342,413	8	2,820,309	0.002	0.191 (0.095, 0.384)		0.247 (0.122, 0.498)	
per 1 kg/m^2^					0.897 (0.878, 0.917)	<0.0001	0.872 (0.851, 0.893)	<0.0001
**Oropharynx cancer**								
<18.5	357,470	77	2,888,072	0.026	1.271 (1.003, 1.609)	0.0173	1.584 (1.25, 2.008)	0.0002
18.5–23	3,750,066	648	30,828,264	0.021	1 (Ref.)		1 (Ref.)	
23–25	2,358,962	481	19,451,513	0.024	1.176 (1.045, 1.323)		0.941 (0.836, 1.06)	
25–30	2,789,174	553	23,002,036	0.024	1.143 (1.021, 1.281)		0.900 (0.802, 1.011)	
≥30	342,413	55	2,820,158	0.019	0.928 (0.705, 1.223)		0.988 (0.748, 1.306)	
per 1 kg/m^2^					1.002 (0.998, 1.005)	0.4004	0.977 (0.961, 0.993)	0.0044
**Oral cavity cancer**								
<18.5	357,470	78	2,888,052	0.027	1.044 (0.828, 1.318)	0.0224	1.277 (1.011, 1.611)	0.0637
18.5–23	3,750,066	799	30,827,875	0.025	1 (Ref.)		1 (Ref.)	
23–25	2,358,962	559	19,451,367	0.028	1.108 (0.995, 1.235)		0.918 (0.823, 1.023)	
25–30	2,789,174	708	23,001,829	0.030	1.187 (1.073, 1.313)		0.947 (0.854, 1.05)	
≥30	342,413	81	2,820,124	0.028	1.109 (0.882, 1.393)		1.036 (0.822, 1.306)	
per 1 kg/m^2^					1.002 (1.000, 1.004)	0.0374	0.988 (0.974, 1.002)	0.0915
**Nasopharynx cancer**								
<18.5	357,470	26	2,888,168	0.009	0.735 (0.494, 1.093)	0.0003	0.917 (0.616, 1.364)	0.8224
18.5–23	3,750,066	379	30,828,872	0.012	1 (Ref.)		1 (Ref.)	
23–25	2,358,962	290	19,451,960	0.014	1.212 (1.04, 1.412)		1.009 (0.865, 1.177)	
25–30	2,789,174	372	23,002,618	0.016	1.314 (1.139, 1.516)		1.077 (0.93, 1.246)	
≥30	342,413	34	2,820,205	0.012	0.980 (0.69, 1.391)		0.986 (0.692, 1.406)	
per 1 kg/m^2^					1.002 (1.001, 1.004)	0.0103	1.002 (0.999, 1.005)	0.2045
**Salivary gland cancer**								
<18.5	357,470	42	2,888,120	0.014	1.051 (0.765, 1.443)	0.0013	1.26 (0.918, 1.731)	0.3857
18.5–23	3,750,066	427	30,828,728	0.013	1 (Ref.)		1 (Ref.)	
23–25	2,358,962	342	19,451,850	0.017	1.269 (1.101, 1.463)		1.094 (0.948, 1.262)	
25–30	2,789,174	410	23,002,358	0.017	1.287 (1.124, 1.473)		1.073 (0.934, 1.232)	
≥30	342,413	52	2,820,164	0.018	1.331 (0.998, 1.775)		1.229 (0.918, 1.645)	
per 1 kg/m^2^					1.002 (1.000, 1.004)	0.0148	1.002 (0.998, 1.006)	0.4104

Model 1: Unadjusted; Model 2: Age, sex, income, smoking, alcohol consumption, regular exercise, diabetes, and hypertension.

**Table 3 cancers-14-03880-t003:** Hazard ratios of head and neck cancer and its subtypes according to waist circumference level.

			Per 1000 People-Year					
Waist Circumference	N	Event	Duration	Incidence Rate	Model 1	*p*-Value	Model 2	*p*-Value
**Head and neck cancer**								
<80/75	3,540,028	2934	29,181,828	0.100	0.643 (0.609, 0.678)	<0.0001	1.104 (1.045, 1.165)	0.0118
<85/80	2,275,478	2637	18,738,360	0.140	0.900 (0.851, 0.950)		1.027 (0.972, 1.085)	
<90/85	1,897,825	2443	15,606,650	0.156	1 (Ref.)		1 (Ref.)	
<95/90	1,112,798	1640	9,128,684	0.179	1.148 (1.078, 1.222)		1.038 (0.975, 1.105)	
<100/95	497,320	720	4,069,838	0.176	1.130 (1.040, 1.228)		1.019 (0.938, 1.108)	
≥100/95	274,636	358	2,237,532.57	0.160	1.022 (0.915, 1.142)		0.998 (0.893, 1.116)	
per 1 kg/m^2^					1.007 (1.007, 1.008)	<.0001	0.996 (0.993, 0.998)	0.0005
**Larynx cancer**								
<80/75	3,540,028	776	29,188,005	0.026	0.619 (0.558, 0.687)	<.0001	1.118 (1.007, 1.241)	0.2644
<85/80	2,275,478	753	18,743,817	0.040	0.935 (0.843, 1.038)		1.071 (0.965, 1.189)	
<90/85	1,897,825	671	15,611,972	0.043	1 (Ref.)		1 (Ref.)	
<95/90	1,112,798	488	9,132,077	0.053	1.244 (1.107, 1.397)		1.104 (0.982, 1.241)	
<100/95	497,320	197	4,071,362	0.048	1.125 (0.96, 1.318)		1.028 (0.877, 1.206)	
≥100/95	274,636	87	2,238,307	0.038	0.904 (0.723, 1.13)		0.945 (0.755, 1.182)	
per 1 kg/m^2^					1.008 (1.007, 1.008)	<0.0001	0.995 (0.990, 1.000)	0.0374
**Sinus cancer**								
<80/75	3,540,028	146	29,190,044	0.005	0.705 (0.551, 0.902)	<0.0001	1.151 (0.895, 1.480)	0.3409
<85/80	2,275,478	126	18,745,985	0.006	0.946 (0.733, 1.221)		1.086 (0.841, 1.403)	
<90/85	1,897,825	111	15,613,882	0.007	1 (Ref.)		1 (Ref.)	
<95/90	1,112,798	86	9,133,546	0.009	1.324 (0.999, 1.754)		1.197 (0.903, 1.587)	
<100/95	497,320	49	4,071,962	0.012	1.691 (1.208, 2.367)		1.477 (1.053, 2.071)	
≥100/95	274,636	21	2,238,589	0.009	1.318 (0.827, 2.102)		1.193 (0.746, 1.908)	
per 1 kg/m^2^					1.007 (1.005, 1.009)	<0.0001	1.003 (0.994, 1.011)	0.5289
**Hypopharynx cancer**								
<80/75	3,540,028	288	29,189,812	0.009	0.696 (0.584, 0.829)	0.0004	1.268 (1.061, 1.514)	<0.0001
<85/80	2,275,478	250	18,745,729	0.013	0.939 (0.784, 1.125)		1.088 (0.908, 1.305)	
<90/85	1,897,825	222	15,613,712	0.014	1 (Ref.)		1 (Ref.)	
<95/90	1,112,798	101	9,133,539	0.011	0.777 (0.614, 0.983)		0.679 (0.537, 0.86)	
<100/95	497,320	49	4,071,962	0.012	0.845 (0.62, 1.151)		0.749 (0.549, 1.021)	
≥100/95	274,636	19	2,238,618	0.008	0.596 (0.373, 0.952)		0.593 (0.37, 0.949)	
per 1 kg/m^2^					1.007 (1.005, 1.009)	<0.0001	0.973 (0.965, 0.981)	<0.0001
**Oropharynx cancer**								
<80/75	3,540,028	477	29,189,087	0.016	0.575 (0.505, 0.654)	<0.0001	0.982 (0.861, 1.119)	0.6138
<85/80	2,275,478	431	18,745,075	0.023	0.809 (0.709, 0.924)		0.909 (0.796, 1.038)	
<90/85	1,897,825	444	15,612,773	0.028	1 (Ref.)		1 (Ref.)	
<95/90	1,112,798	288	9,132,911	0.031	1.109 (0.956, 1.287)		1.017 (0.877, 1.18)	
<100/95	497,320	121	4,071,721	0.029	1.045 (0.855, 1.278)		0.982 (0.803, 1.202)	
≥100/95	274,636	53	2,238,476	0.023	0.834 (0.627, 1.108)		0.874 (0.657, 1.164)	
per 1 kg/m^2^					1.008 (1.007, 1.009)	<0.0001	1.000 (0.994, 1.006)	0.9762
**Oral cavity cancer**								
<80/75	3,540,028	602	29,188,727	0.020	0.643 (0.571, 0.724)	<0.0001	1.032 (0.915, 1.165)	0.8636
<85/80	2,275,478	559	18,744,814	0.029	0.930 (0.824, 1.049)		1.066 (0.945, 1.203)	
<90/85	1,897,825	501	15,612,741	0.032	1 (Ref.)		1 (Ref.)	
<95/90	1,112,798	319	9,132,863	0.034	1.088 (0.946, 1.253)		0.988 (0.859, 1.138)	
<100/95	497,320	158	4,071,647	0.038	1.210 (1.012, 1.447)		1.054 (0.881, 1.262)	
≥100/95	274,636	86	2,238,454	0.038	1.199 (0.954, 1.507)		1.075 (0.854, 1.354)	
per 1 kg/m^2^					1.007 (1.006, 1.008)	<0.0001	0.999 (0.994, 1.004)	0.6360
**Nasopharynx cancer**								
<80/75	3,540,028	308	29,189,410	0.010	0.655 (0.555, 0.774)	<0.0001	0.958 (0.808, 1.135)	0.1366
<85/80	2,275,478	251	18,745,489	0.013	0.830 (0.697, 0.989)		0.893 (0.749, 1.064)	
<90/85	1,897,825	252	15,613,350	0.016	1 (Ref.)		1 (Ref.)	
<95/90	1,112,798	178	9,133,207	0.019	1.207 (0.997, 1.463)		1.153 (0.951, 1.397)	
<100/95	497,320	73	4,071,856	0.017	1.109 (0.855, 1.439)		1.092 (0.841, 1.419)	
≥100/95	274,636	39	2,238,511	0.017	1.078 (0.769, 1.510)		1.162 (0.827, 1.632)	
per 1 kg/m^2^					1.007 (1.006, 1.009)	<0.0001	1.004 (1.000, 1.008)	0.0372
**Salivary gland cancer**								
<80/75	3,540,028	373	29,189,164	0.012	0.744 (0.636, 0.871)	<0.0001	1.055 (0.898, 1.240)	0.2357
<85/80	2,275,478	293	18,745,356	0.015	0.911 (0.772, 1.075)		1.014 (0.859, 1.198)	
<90/85	1,897,825	268	15,613,314	0.017	1 (Ref.)		1 (Ref.)	
<95/90	1,112,798	198	9,133,084	0.021	1.263 (1.051, 1.518)		1.176 (0.978, 1.413)	
<100/95	497,320	84	4,071,832	0.020	1.202 (0.941, 1.536)		1.077 (0.842, 1.378)	
≥100/95	274,636	57	2,238,470	0.025	1.484 (1.115, 1.975)		1.355 (1.016, 1.808)	
per 1 kg/m^2^					1.007 (1.005, 1.008)	<0.0001	1.004 (0.999, 1.008)	0.0977

Model 1: Unadjusted; Model 2: Age, sex, income, smoking, alcohol consumption, regular exercise, diabetes, and hypertension.

**Table 4 cancers-14-03880-t004:** Interactions analysis: body mass index with smoking on the occurrence of head and neck cancer.

Smoking Status	BMI	N	Event	Duration	Incidence Rate	Hazard Ratio	*p*-Value	*p* for Interaction
**Never smoker**	<18.5	254,354	138	2,072,506	0.067	1.219 (1.024, 1.451)	0.0091	
	18.5–23	2,427,070	1531	20,025,284	0.076	1 (Ref.)		
	23–25	1,359,999	1117	11,241,965	0.099	0.958 (0.887, 1.035)		
	25–30	1,533,879	1321	12,674,257	0.104	0.926 (0.859, 0.999)		
	≥30	197,060	166	1,625,840	0.102	1.090 (0.927, 1.282)		
**Ever smoker**	<18.5	103,116	276	814,765	0.338	1.335 (1.178, 1.512)	<0.0001	
	18.5–23	1,322,996	2509	10,793,028	0.232	1 (Ref.)		
	23–25	998,963	1676	8,202,268	0.204	0.872 (0.819, 0.928)		
	25–30	1,255,295	1853	10,319,484	0.179	0.817 (0.768, 0.869)		
	≥30	145,353	145	1,193,494	0.121	0.763 (0.644, 0.903)		

Adjusted for age, sex, income, smoking, alcohol consumption, regular exercise, diabetes, and hypertension.

**Table 5 cancers-14-03880-t005:** Interactions analysis: body mass index with drinking on the occurrence of head and neck cancer.

Drinking Status	BMI	N	Event	Duration	Incidence Rate	Hazard Ratio	*p*-Value	*p* for Interaction
**Never drinker**	<18.5	202,657	182	1,627,124	0.111	1.193 (1.024, 1.390)	0.0002	
	18.5–23	2,001,560	1781	16,423,432	0.108	1 (Ref.)		
	23–25	1,186,970	1206	9,771,319	0.123	0.918 (0.853, 0.988)		
	25–30	1,370,271	1394	11,285,571	0.123	0.884 (0.823, 0.949)		
	≥30	177,853	163	1,463,710	0.111	0.994 (0.845, 1.170)		
**Ever drinker**	<18.5	154,813	232	1,260,147	0.184	1.448 (1.265, 1.659)	<0.0001	
	18.5–23	1,748,506	2259	14,394,880	0.156	1 (Ref.)		
	23–25	1,171,992	1587	9,672,914	0.164	0.881 (0.826, 0.940)		
	25–30	1,418,903	1780	11,708,169	0.152	0.820 (0.769, 0.873)		
	≥30	164,560	148	1,355,624	0.109	0.792 (0.670, 0.938)		

Adjusted for age, sex, income, smoking, alcohol consumption, regular exercise, diabetes, and hypertension.

## Data Availability

Data available on request due to data sharing restrictions. The data presented in this study are available on request from the corresponding author.

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
