# Peer review of "Associations of Body Mass Index and Waist Circumference with the Risk of Head and Neck Cancer: A National Population-Based Study"

_cancers, 2022, doi:10.3390/cancers14163880_

Round 1
Reviewer 1 Report
Good work.
A minor revisions are necessary: in the literature is described the role of some neuropeptides connected to satiety and food intake. For our opinion, some of these manuscript must be mentioned in your work.
For example in the chapter "Sireci F, Cappello F, Canevari FR, Dispenza F, Gallina S, Salvago P, Martines F. The role of Leptin in laryngeal squamous cell carcinoma. In: Évariste Gilles and Daniel Mickaël eds. Leptin: Production, Regulation and Functions, 2017, chapter 6, pp. 167–176" the autohors evidenced high expression of leptin (a neuropeptide related with the body mass index) in laryngeal cancer.
Author Response
Response to Reviewer 1 Comments
A minor revisions are necessary: in the literature is described the role of some neuropeptides connected to satiety and food intake. For our opinion, some of these manuscript must be mentioned in your work.
1.For example in the chapter "Sireci F, Cappello F, Canevari FR, Dispenza F, Gallina S, Salvago P, Martines F. The role of Leptin in laryngeal squamous cell carcinoma. In: Évariste Gilles and Daniel Mickaël eds. Leptin: Production, Regulation and Functions, 2017, chapter 6, pp. 167–176" the autohors evidenced high expression of leptin (a neuropeptide related with the body mass index) in laryngeal cancer.
Answer: we agree with review’s opinion. Some neuropeptides linked to smoking ,drinking, body weight and cancer develop. Therefore,
Following sentence was inserted in the discussion.
“Some recent studies suggested that neuropeptide such as leptin was influenced by smoking and drinking and affect body weight by regulating satiety and food intake [17-21]. Gallina reported that leptin was associated with recurrence of malignancy in laryngeal cancer [21].”
We have added these references.
- Gonseth S, Locatelli I, Bize R et al. Leptin and smoking cessation: secondary analyses of a randomized controlled trial assessing physical activity as an aid for smoking cessation. BMC Public Health 2014;14:911
- Shen L, Cordero JF, Wang JS et al. The effect of leptin on blood pressure considering smoking status: a Mendelian randomization study. Hypertens Res 2020;43:342-9.
- Weinland C, Tanovska P, Kornhuber J, Mühle C, Lenz B. Serum lipids, leptin, and soluble leptin receptor in alcohol dependence: A cross-sectional and longitudinal study. Drug Alcohol Depend 2020;209:107898.
- Pan WW, Myers MG, Jr. Leptin and the maintenance of elevated body weight. Nat Rev Neurosci 2018;19:95-105.
- Gallina S, Sireci F, Lorusso F, DI Benedetto DV, Speciale R, Marchese D, et al. The immunohistochemical peptidergic expression of leptin is associated with recurrence of malignancy in laryngeal squamous cell carcinoma. Acta Otorhinolaryngol Ital. 2015;35:15-22.

Reviewer 2 Report
A major revision is required, and after responding to the following remarks and revising the paper, the manuscript may be considered for publication.
1.The title may lead to misunderstanding. I suggest changing the title to "Associations of body mass index and waist circumference with the risk of head and neck cancer: A national population-based study".
2.BMI and WC have been found to be associated with a lot of cancers; the authors should fully reveal the unique relations between BMI/WC and HNC in the introduction section.
3.There are massive tables in the manuscript which are too complicated to read. Also, readers can hardly get the main point. Figures are needed to demonstrate the data and highlight the most significant findings.
4.All tables except table 1 should be refined and shorted.
5.Please uniform the decimal digits in the whole passage.
6.On page15, the data in the results section conflicts with that in table6. Double check the original data and correct the corresponding mistakes.
7.Also,check the typos throughout the manuscript during revision submission.
8.How has the follow-up been done? The BMI/WC of each individual been determined on a single medical examination? Provide more detailed information about follow-ups.
9.The study period is quite long, and An individual is likely to change their lifestyle. Such bias must be considered, and more subgroups should be introduced to control confounders.
10.The development and progression of cancer is a complicated long-term process; the causal relationship between changes in BMI/WC and cancer development should be fully discussed. Whether there may be an ecological fallacy?
11.More limitations should be discussed. Such as this is a retrospective study etc.
Author Response
Response to Reviewer 2 Comments
A major revision is required, and after responding to the following remarks and revising the paper, the manuscript may be considered for publication.
1.The title may lead to misunderstanding. I suggest changing the title to "Associations of body mass index and waist circumference with the risk of head and neck cancer: A national population-based study".
Answer: The word “cancer” was inserted in the title.
2.BMI and WC have been found to be associated with a lot of cancers; the authors should fully reveal the unique relations between BMI/WC and HNC in the introduction section.
Answer: BMI has been known to be positively associated with multiple cancer type. However. in head and neck cancer (HNC) the relationship between BMI and HNC incidence is complicated. This is because smoking and drinking not only affect the incidence of HNC, but also affect BMI/WC
These contents are described as below in the introduction.
“Some subtypes of HNC caused by smoking and alcohol can affect BMI by causing dysphagia [5]”
“After analyzing the data of 17 case-control studies that included 12,716 cases and 17,439 controls, Gaudet et al. suggested that a low BMI was associated with a higher risk of HNC, regardless of smoking and drinking [5].”
Therefore, the purposes of this study are:
1.We investigated the association between BMI and the incidence of HNC subtypes and also the relationship between WC and the HNC subtype
2.We performed stratified analysis about smoking and drinking in the occurrence of HNC according to BMI and WC category
3.There are massive tables in the manuscript which are too complicated to read. Also, readers can hardly get the main point. Figures are needed to demonstrate the data and highlight the most significant findings.
Answer: we added figure 1. That presented the hazard ratio of the HNC subtype according to the BMI category, which is the core of this manuscript
4.All tables except table 1 should be refined and shorted.
Answer: according to reviewer’s opinion
1.In Table 2 and Table3: model 2 was deleted and replaced with model 3
We simplified the existing 3 groups (Model 1.2.and 3) to 2 groups (Model 1 and 2).
-Model 1: was not adjusted
-Model 2: previously model 3, variables that could affect the incidence of HNC or influence of BMI and WC were adjusted.
In the manuscript, 2.4 statistical analysis section was also modified as follows.
“Model 1 was not adjusted; Model 2 was adjusted for age, sex, income, smoking, alcohol consumption, exercise and chronic diseases that could affect BMI or WC, such as hypertension and diabetes. In Model 2, variables that could affect the incidence of HNC or influence of BMI and WC were adjusted”
- For interaction test
Table 4 and table 5 were shorted:
-Table 4. and Table 5. showed the effect of smoking or drinking status on the risk of overall HNC according to BMI category.
Table 6 as well as the paragraph for interaction for the BMI category and sex were deleted
-The number of female with HNC onset according to the BMI category was very different from the number of male
5.Please uniform the decimal digits in the whole passage.
Answer: The decimal point of “Duration” has been deleted from the entire table. Except p- values( 4 digits), remaining values (Incidence rate and Hazard ratio) were unified to 3 decimal digits.
6.On page15, the data in the results section conflicts with that in table6. Double check the original data and correct the corresponding mistakes.
Answer: We collected according to the table. However, the number of female with HNC onset according to the BMI category was very different from the number of male and the gender difference were not related to the purpose of this study. We decide to remove the interaction test for BMI and sex
7.Also,check the typos throughout the manuscript during revision submission.
Answer: Throughout the manuscript, we checked the typos and revised
8.How has the follow-up been done? The BMI/WC of each individual been determined on a single medical examination? Provide more detailed information about follow-ups.
Answer: We selected subjects who had undergone single health checkups including BMI and WC in 2009. We monitored subjects until 31 December 2018 with median follow-up duration of 8.31(8.12–8.57) years. Participants were censored if they (1) died, (2) immigrated and were no longer followed up by the NHIS, or (3) had new-onset HNC (primary outcome of the study).
9.The study period is quite long, and An individual is likely to change their lifestyle. Such bias must be considered, and more subgroups should be introduced to control confounders.
Answer: We agree review’s opinion that lifestyle could be changed during follow up period.
This is one of the disadvantages of 10-year retrospective cohort study. We included this in the limitation of the manuscript.
“Finally, this study was designed as 10 years retrospective cohort study. The lifestyle (including smoking or drinking status) of participants could be changed during observation period, however it is difficult to reflect these changes.”
10.The development and progression of cancer is a complicated long-term process; the causal relationship between changes in BMI/WC and cancer development should be fully discussed. Whether there may be an ecological fallacy?
à Answer: We agree review’s opinion that BMI/WC could be changed before and during cancer developed. It is best to know the BMI/WC of the participants prior to the onset of cancer, but this is practically impossible in large observational studies. Therefore, to minimize the effect of cancer on BMI/ WC, Subjects who diagnosed with cancer within one year of the health check-up were excluded. This study was retrospective cohort study, we did not check increasing or decreasing BMI/WC. Only observed BMI/WC status and HNC risk. Therefore, the causal relationship between changes in BMI/WC and HNC development would need well designed case control study and beyond the scope of this study. We added this limitation of the study
11.More limitations should be discussed. Such as this is a retrospective study etc.
à Answer: The concerns about 8-10 question above were added as limitations of this study
Reviewer 3 Report
This is a well conceptualized and reported manuscript, although not entirely novel, specifically provides data regarding the role of BMI and waist circumference denoting abdominal body fat distribution and risk of head and neck cancers. The confounders of gender, age, smoking and alcohol use have been taken into account. The sample is sizeable and well annotated derived from the Korean national health registry. The methods, the results and discussion are precisely stated, compared to other national registry data from previous years.
Addition of the results pertaining to alcohol and smoking data to the abstract is recommended.
The authors should also add the rationale for including and examining waist circumference and the potential mechanism that formed the bassis of their inclusion of this variable.
A major limitation of this study is that the variable of nutritional intake and composition of nutritional intake was not accounted for. This should be noted in their last paragraph describing the limitations of the study as nutritional intake is a major contributor of body composition.
Overall these are minor edits to a well reported and important manuscript.
Author Response
Response to Reviewer 3 Comments
This is a well conceptualized and reported manuscript, although not entirely novel, specifically provides data regarding the role of BMI and waist circumference denoting abdominal body fat distribution and risk of head and neck cancers. The confounders of gender, age, smoking and alcohol use have been taken into account. The sample is sizeable and well annotated derived from the Korean national health registry. The methods, the results and discussion are precisely stated, compared to other national registry data from previous years.
Addition of the results pertaining to alcohol and smoking data to the abstract is recommended.
Answer : According to the reviewer’s comments, we have added this sentence in the abstract. “The HR for occurrence of HNC was high in participants with underweight according to smoking status (1.219 for never smoker vs 1.448 for ever smoker, p for interaction=0.0015) and drinking status (1.193 for never drinker vs 1.448 for ever drinker, p for interaction=0.0044).
The authors should also add the rationale for including and examining waist circumference and the potential mechanism that formed the basis of their inclusion of this variable.
Answer : According to the reviewer’s comments, we have added this sentence in the discussion.
“Abdominal obesity, rather than general obesity, is considered to be one of the key features of metabolic syndrome [36]. Intra-abdominal fat has been hypothesised to be biologically different from fat in other areas with regard to tumor angiogenesis and cell proliferation [37].”
We have added these references.
Bruce KD, Byrne CD. The metabolic syndrome: common origins of a multifactorial disorder. Postgrad Med 2009;85:614–621.
Klopp AH, Zhang Y, Solley T, Amaya-Manzanares F, Marini F, Andreeff M, et al. Omental adipose tissue-derived stromal cells promote vascularization and growth of endometrial tumors. Clin. Cancer Res. 2012;18:771–782.
A major limitation of this study is that the variable of nutritional intake and composition of nutritional intake was not accounted for. This should be noted in their last paragraph describing the limitations of the study as nutritional intake is a major contributor of body composition.
Answer : According to the reviewer’s comments, we have added this sentence in the discussion.
“Although there are some case-control studies about the association between nutrients and HNC, this study did not contain detailed individual nutritional status or diet. Nutritional intake is one of the leading contributing factors to obesity. Future studies should include a more accurate and comprehensive dietary assessment to determine the role of diet and body composition in HNC development.”
Overall these are minor edits to a well reported and important manuscript.

Round 2
Reviewer 2 Report
The manuscript has been greatly improved. I think it can be published in present form.